# Regional differences in agricultural and socioeconomic factors associated with farmer household dietary diversity in India

**Sukhwinder Singh**[1]*, **Andrew D. Jones**[2], **Meha Jain**[3]

**1** Center for Chronic Disease Control, Public Health Foundation of India, Gurgaon, India, **2** School of Public Health, University of Michigan, Ann Arbor, MI, United States of America, **3** School for Environmental and Sustainability (SEAS), University of Michigan, Ann Arbor, MI, United States of America

* sukhrec@yahoo.com

**Data Availability Statement:** All relevant data are within the paper and its Supporting Information files.

## Abstract

This paper investigated whether there are any regional-level differences in factors associated with farmer household dietary diversity using the Food Consumption Score (FCS), in two states of India: Haryana and Gujarat. Our results suggest that the factors associated with farmer household dietary diversity were region-specific, with diverse drivers across districts. For example, in Vadodara (Gujarat), farmers who had greater crop diversity and planted more cash crops had higher dietary diversity while large landholders in Bhavnagar (Gujarat) had higher dietary diversity. In Karnal (Haryana), more educated farmer households and those who cultivated large landholdings had higher dietary diversity while farmers in Bhiwani (Haryana) who were more educated and sold more crops to market had higher dietary diversity. Thus, factors associated with FCS differed even within the same state. These results suggest that in some regions of India, higher crop diversity and better education could improve farmer household dietary diversity. On the other hand, in some other regions, dietary diversity is best improved through the income generation pathway, where households that earn increased income from selling more crops were able to purchase more diverse food from markets. Our study suggests that the drivers of household dietary diversity across rural India are complex and heterogeneous; thus, future policies and programs to improve farmer household nutrition should be tailored considering regional differences in the factors associated with household nutrition.

## 1. Introduction

Food insecurity or malnutrition is a global burden, with one in three people affected worldwide. In fact, every country in the world is facing at least one form of malnutrition, while 123 countries are facing a "triple burden" of energy and micronutrient deficiencies along with rising rates of obesity [1,2,3]. Malnutrition is especially high in Southern Asia where about 281 million people are undernourished (i.e. they have inadequate calorie intake relative to their nutritional needs). Nearly one-third of children in India are stunted, wasted, or underweight while one-fifth of adults are overweight or obese. Further, one-fifth of men, and half of children and women are anemic [4,5]. Reducing rural malnutrition, particularly child

**Funding:** Specific grant numbers: IMMANA 169864-2013 Initials of authors who received each award: Sukhwinder Singh Funded the study: The UK Department for International Development (DFID) Initials of authors who received salary or other funding from commercial companies: Sukhwinder Singh URLs to sponsors websites: https://immana.lcirah.ac.uk The funders had no role in study design, data collection and analysis, decision to publish, or preparation of the manuscript.

**Competing interests:** We declare that no competing interests exist.

malnutrition, has been a main policy goal over the last decade [2]. For example, the Indian government passed the National Food Security Act (NFSA) in 2013 to provide subsidized food grains to poorer sections of society in an attempt to reduce malnutrition rates [6] The impacts of these efforts on malnutrition, however, are yet to be investigated [7]. In order for such policies to be effective, it is critical that they consider the socioeconomic and agricultural factors associated with improved household nutrition across regions.

Many studies have examined the agricultural (e.g. agricultural biodiversity, farm income) and socioeconomic (e.g. family income, educational status, market integration) factors associated with farmer household dietary diversity, one of the key indicators of household nutrition. Jones [8], and Sibhatu and Qaim [9] did a meta-analysis of these studies and concluded that agricultural biodiversity has a clear and consistent association with farmer household-and individual dietary diversity in low- and medium- income countries although the magnitude of this association was small. Further, Kumar et al. [10] and Das [11] showed that agricultural production diversity and education attainment, respectively, are some of the major determinants of household dietary diversity in India. Some other studies [12,13,14] reported that higher farm incomes are significantly associated with improvements in farmer household dietary diversity. Similarly, socioeconomic drivers such as educational status and market integration were found to be more influential drivers of household dietary diversity than agricultural biodiversity in Indonesia, Kenya, Ethiopia, and Malawi [15,16].

While many studies have examined the agricultural drivers and socio-economic of farm household dietary diversity, few studies have identified the extent to which these drivers may vary across regions within a given country or state. Yet, doing so is important given that the studies that have examined this question have found regional differences in the drivers of dietary diversity. For example, Demeke et al. [17] and Ochieng et al. [18] found regional-level differences in the factors associated with household dietary diversity in Kenya and Tanzania, respectively. In addition, Parappurathu et al. [19] investigated food consumption patterns and dietary diversity of farmers at the village-level across twelve villages in India and found a significant difference in dietary diversity across these villages, though this study did not consider the drivers of dietary diversity. These studies suggest that there is likely not a universal pattern in which factors influence dietary diversity and instead, the drivers of dietary diversity are likely to be diverse and differ across regions.

To better understand how heterogeneous the drivers of rural household nutrition are across India, this study investigated the association between agricultural and socioeconomic factors and farmer household dietary diversity across districts in Gujarat and Haryana, two Indian states with different crop production and food consumption patterns. We aim to understand how agricultural and socioeconomic factors are associated with household dietary diversity, and how these associations vary across districts in both states. Specifically, this study investigates which agricultural and socioeconomic factors are associated with farmer household dietary diversity across districts within Gujarat and Haryana. This study has important implications for regional policies that aim to improve farmer household dietary diversity in rural India. Our work may help policy makers understand the regional-level differences in the drivers of household dietary diversity in rural India, allowing for more tailor-made policies and programs to improve household nutrition.

## 2. Material and methods

### 2.1. Study locations and sampling methodology

We selected two Indian states, Gujarat and Haryana (Fig 1), because these are the two states in India where over 70% of rural households depend on agriculture and they vary significantly

considering crop and income diversity ([20,21,22]; www.esaharyana.gov.in; www.dag.gujarat.gov.in). For instance, the average crop diversity in Gujarat was 0.73 compared with 0.70 in Haryana. Gujarat, being a large state with a low availability of water for irrigation, could realize cropping intensity of 57% only as compared to 133% in Haryana which is a small state but with plenty of water available for irrigation (S1 and S2 Tables). Further, these states were also quite divergent in terms of per capita livestock (0.21 v/s 0.36) and poultry (0.31 v/s 1.89%). Thus, these states made an ideal case for a comparative study.

We sampled farmers across a gradient of low to high crop, farm, and income diversity because we aimed to examine how agricultural and socioeconomic factors are associated with farmer household dietary diversity. Therefore, we used secondary data on crops grown, livestock owned, and income generated from Indian census statistics. For that, we constructed a 'Farming Intensity Index' (FII: adapted from [23,24,25]) for each district in Gujarat (S1 Table) and Haryana (S2 Table). This paper is based on the same field survey that I, as the first author, used in my paper Singh et al. [23]. Thus, some contents of this paper, e.g. Material and Methods, and Descriptive and Summary Statistics may be similar to Singh et al. [23]. We calculated FII using four major indicators of crop diversity (e.g. Crop Diversity Index-CDI: Section 2.4), farm diversity (e.g. per capita livestock, and per capita poultry population), and farm income (e.g. total cropped area as percent of total land area). Further, since previous literature confirms a strong relationship between household education and farmer household dietary diversity [26,27,28], we included rural literacy rates, as a factor in FII. Then, we standardized these figures and took the average of our measures of crop and income diversity to weight each of these equally. We then summed these weighted values to obtain FII values using the following formula:

$$FII = \sum_{i=1}^{n} \frac{X - \bar{x}}{SD}$$

where.

X = observed values of variable of interest,

$\bar{x}$ = Mean of X,

SD = Standard Deviation of X,

n = Number of variables.

Based on these FII values, three districts were selected in each state (Fig 1): One district that had an FII value close to the state average, one district that had one of the two highest FIIs within each state, and one district that had one of the two lowest FII values within each state. To ensure that the selected districts were spread across the state and were not next to one another, we also considered spatial location of these districts (methods are the same as those from Singh et al. [23]).

To maintain consistency, the same formula and methodology was used to select blocks within each of these three districts in Gujarat and Haryana (S3 and S4 Tables). Due to the unavailability of village-level data, we randomly selected three sets of villages within each block while in the field. These village sets were: one set that was close to a city, one set that was close to a highway, and one set that was far from both a city and a highway (Fig 2). This sampling strategy ensured the inclusion of villages with varied income diversity because rural households in villages that are closer to cities and highways are more likely to take part in non-agricultural livelihoods, e.g. salaried professions or running a business. Two to three adjoining villages formed each village set. Using purposive-cum-random sampling, approximately 30 farmer households within each village set were selected. Surveys were conducted by visiting the selected farm households at their home (methods are the same as those from Singh et al. [23]). A farmer

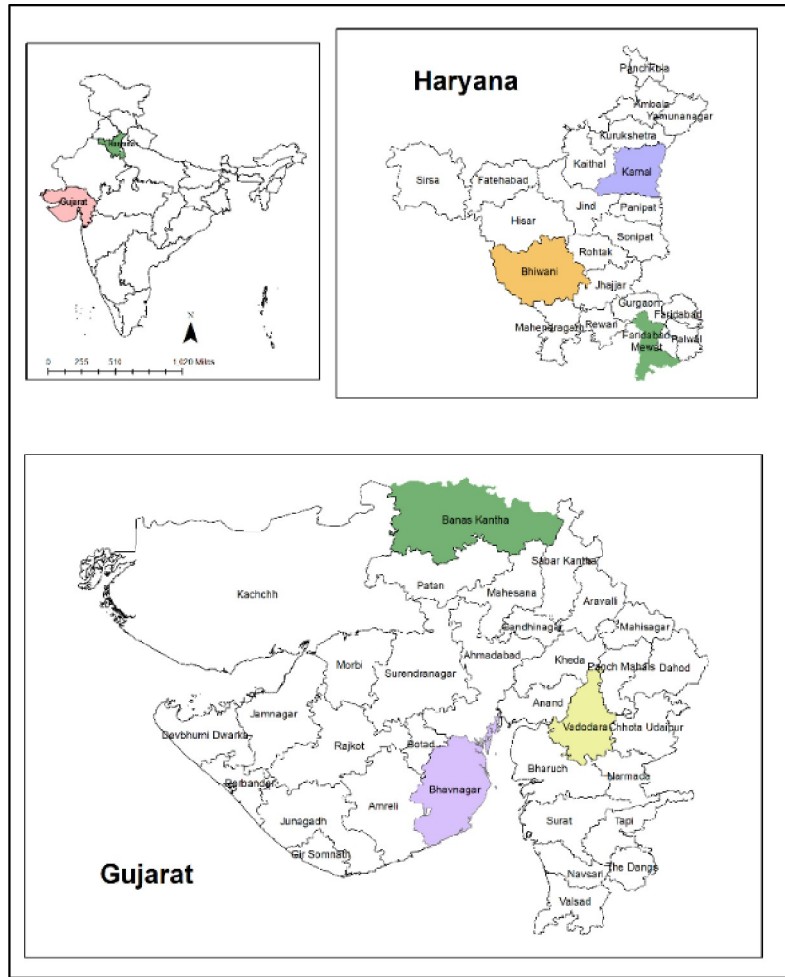

**Fig 1. A map of states, districts and blocks selected for this study in India (Fig from Singh et al. [23]).**

household here refers to a family directly engaged in farming, particularly crop production and not necessarily owning any land. Those who rented or leased land from others were also included. However, those, owning but not cultivating the land themselves were excluded.

## 2.2 Survey data collection

A structured survey schedule was prepared and data were collected using a mobile-based application. Two separate teams of five female enumerators conducted the surveys in each state during May-June 2017. Data related to crops grown, farm-related activities, income sources, demographic information and food consumption were collected from the head of the household, typically a male, and the main preparer of food, typically the wife of the male head of the household. This survey was reviewed and approved by the Institution Review Board, University of Michigan, Ann Arbor, USA (IRB Approval Number: IRB00000246).

## 2.3. Household dietary diversity assessment

Farmer household dietary diversity can be assessed by computing the Food Consumption Score (FCS). The Food Consumption Score (FCS) aggregates household-level data on the

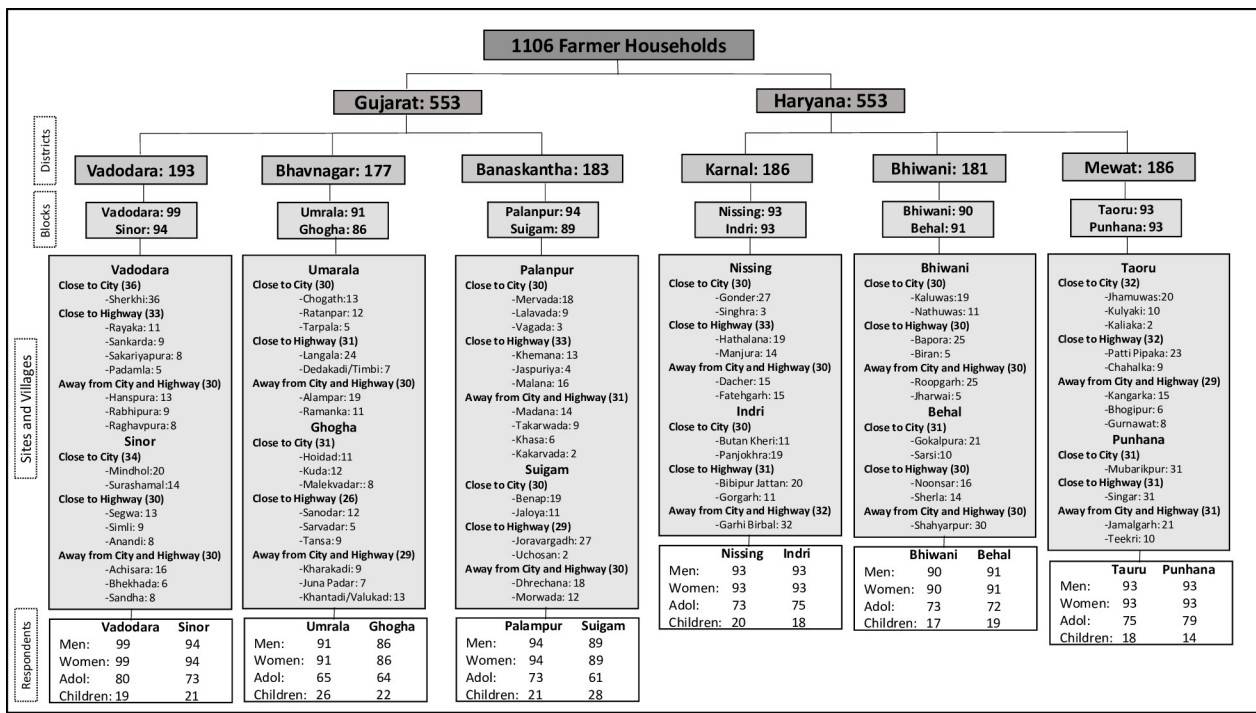

**Fig 2. Sample distribution showing number of farmer households surveyed in each district and block in Gujarat and Haryana (Fig from Singh et al. [23]).**

diversity and frequency of nine food groups (e.g. Main staples, Pulses, Vegetables, Fruit, Meat and Fish, Milk, Sugar, Oil, and Condiments) consumed over the past week, which is then weighted according to each group's relative nutritional value. FCS can be further classified as poor, borderline, or acceptable [29]. FCS is associated with caloric intake [30,31], but has not been validated for nutrient adequacy [32].

## 2.4. Metrics constructed

**2.4.1. Crop diversification index.** The Crop Diversification Index (CDI) was calculated for each farmer household surveyed using the 1-*H* formula, where *H* is Hirschman-Herfindahl Index (HHI) measured as:

$$H = \sum_{i=1}^{N} s_i^2$$

where.

*N* is the total number of crops during 2016–17,

*Si* represents area proportion of the i-th crop in total cropped area.

*H* takes a value of 1 when there is a monoculture and approaches zero with increasing diversity. Therefore, when using 1-*H*, a larger number indicates greater crop diversity [24,23]. The CDI was calculated using all crops grown during the whole year.

**2.4.2. Family education index.** The Family Education Index (FEI) for each farmer household was calculated by adding the education level of all adults and adolescents in a household and dividing the resulting value by the number of all adults and adolescents.

**2.4.3. Food consumption score.** The Food Consumption Score (FCS) is the main measure of dietary quality at the household level recommended by the World Food Program [33]. As defined by the WFP [33], the consumption frequency of each of the food groups consumed were multiplied by an assigned weight of each food group and the resulting values were added to calculate the final FCS for each of the farmer households:

$$FCS = a_{staple} x_{staple} + a_{pulse} x_{pulse} + a_{veg} x_{veg} + a_{fruit} x_{fruit} + a_{meat} x_{meat} + a_{sugar} x_{sugar} + a_{dairy} x_{dairy} + a_{oil} x_{oil} + a_{condiments} x_{condiments}$$

where.

FCS = Food Consumption Score;

$x_i$ = frequency of food consumption = number of days on which a food group was consumed during the past 7 days; and.

$a_i$ = weight of each food group.

The assigned weights for each of the nine food groups were based on the energy, protein and micronutrient densities of each of the food groups. These assigned weights were: 1. Main staples-2; 2. Pulses-3; 3. Vegetables-1; 4. Fruit-1; 5. Meat and Fish-4; 6. Milk-4; 7. Sugar-0.5, 8. Oil-0.5, and 9. Condiments-0.

## 2.5. Framework to examine associations

We ran regressions to examine the associations between various agricultural and socioeconomic factors and farmer household dietary diversity across six districts of Gujarat and Haryana in India. In these regressions, we used the following major independent variables:

**2.5.1. Crop diversity index (CDI).** Crop diversity in India is decreasing [20,21], which affects farmer household dietary diversity, especially for those who are sustaining homegrown food [9,15,16,34]. Thus, we decided to include CDI, as a factor, in our regressions.

**2.5.2. Landholding size (in ha).** A larger landholding size can improve household dietary diversity by enabling a farmer household to produce more to consume more at home [9,34] as well as to sell more to enhance income to buy more diverse food from market [12,14]. Thus, it could be an important socioeconomic factor affecting dietary diversity in rural India.

**2.5.3. Per capita annual income (PCAI).** As increasing incomes generally result in improved household dietary diversity [12,13,14], per capita family income was included in the regressions.

**2.5.4. Family education index (FEI).** Many studies argue that education is an important socioeconomic factor associated with household dietary diversity [18,26,27,28,35]; therefore, family education was included in the regressions.

**2.5.5. Distance traveled to food markets (Kms).** As Jones et al. [36] and Koppmair et al. [15] suggested market access and higher household dietary diversity are associated with each other; therefore, we included this in our study. We used average distance traveled by farmers to markets for vegetables, fruits, and pulses as a proxy for market access.

**2.5.6. Crops sold (%).** We found that most farmers in Gujarat and Haryana sell some part of their agricultural production to markets (Table 1) and this increased income may improve dietary diversity [12,13,14,37]. Thus, we included the average percent of crops sold to the market for each farmer in our analyses.

## 2.6. Statistical models

The descriptive statistics of all of variables of interest across each district in two states were tabulated to understand the variation in our variables across study sites. SPSS was used to create all descriptive tables. We ran linear regressions for each of the six districts across two states using R Project Software to identify the associations between agricultural (e.g. crop diversity,

**Table 1. Socioeconomic and agricultural profile of farmer households surveyed in Gujarat and Haryana.**

| Parameter/District/State | Vadodara | Bhavnagar | Banas Kantha | Gujarat | Karnal | Bhiwani | Mewat | Haryana | Overall |
|---|---|---|---|---|---|---|---|---|---|
| **Sample Size (*n*)** | 193 | 177 | 183 | 553 | 186 | 181 | 186 | 553 | 1106 |
| **Socioeconomic Profile** | | | | | | | | | |
| Farmer Age (years) | 45 (12.90) | 46 (14.44) | 42 13.46) | 45 (13.66) | 42 (13.81) | 43 (13.60) | 41 (15.05) | 42 (14.19) | 43 (13.97) |
| Average Farmer Household Size (#) | 6.9 (3.04) | 8.0 (3.36) | 6.4 (2.36) | 7.1 (3.01) | 6.1 (2.52) | 6.6 (2.74) | 8.8 (3.78) | 7.2 (3.27) | 7.1 (3.14) |
| Farmer Household Education (per capita) | 7.4 (2.20) | 6.2 (2.21) | 6.0 (2.50) | 6.6 (2.39) | 7.6 (2.25) | 7.7 (2.37) | 5.1 (2.76) | 6.8 (2.74) | 6.6 (2.57) |
| Family Members Engaged in Agriculture (% of total members) | 79 (25.65) | 87 (22.08) | 83 (23.54) | 83 (24.00) | 94 (15.27) | 89 (20.14) | 92 (16.58) | 92 (17.50) | 88 (21.41) |
| Average Farmer Household Annual Income from all sources (in US$) | $2,950 (4028) | $2,336 (2631) | $2,727 (3191) | $2,677 (3354) | $525 (366) | $466 (737) | $551 (606) | $514 (589) | $1,593 (2634) |
| Income from Farming (% of total household income) | 62 (29.48) | 72 (26.13) | 56 (24.72) | 63 (27.65) | 87 (22.04) | 86 (21.84) | 77 (26.84) | 83 (24.12) | 73 (26.86) |
| **Agricultural Profile** | | | | | | | | | |
| Landholding Size (in ha) | 2.9 (3.9) | 3.6 (3.6) | 2.8 (2.6) | 3.1 (3.5) | 2.5 (3.0) | 2.6 (2.2) | 1.3 (1.7) | 2.1 (2.9) | 2.6 (3.0) |
| Cropping Intensity (%) | 125 (55.93) | 109 (38.09) | 133 (72.39) | 122 (58.14) | 205 (25.07) | 195 (17.32) | 202 (33.35) | 200 (26.40) | 162 (59.69) |
| Crop Diversity Index (CDI) | 0.48 (0.24) | 0.37 (0.27) | 0.63 (0.23) | 0.49 (0.27) | 0.56 (0.07) | 0.68 (0.09) | 0.62 (0.11) | 0.62 (0.11) | 0.56 (0.21) |
| Average Crops Sold (%) | 57 (38.41) | 60 (30.33) | 50 (25.69) | 56 (32.29) | 50 (23.98) | 53 (19.64) | 46 (23.95) | 50 (22.79) | 53 (28.08) |
| Distance to Food Markets (kms) | 4.9 (4.45) | 6.4 (6.18) | 7.9 (5.00) | 6.4 (5.37) | 2.6 (2.45) | 3.5 (3.49) | 3.5 (2.28) | 3.2 (2.81) | 4.8 (4.57) |
| Milk kept for domestic use (%) | 39 (28.57) | 52 (29.22) | 26 (25.00) | 37 (29.38) | 62 (34.32) | 70 (31.94) | 68 (31.05) | 67 (32.51) | 54 (34.37) |

Figures in parenthesis represent Standard Deviation (SD) for respective mean values.

landholding size, farm income and crops sold) and socioeconomic (e.g. family income, family education, distance to food markets and consumption of domestically produced milk) factors, and FCS. All continuous independent variables were normalized by the mean and standard deviation to make coefficient values comparable across all independent variables. We calculated variable inflation factors (VIFs) for each regression and found no evidence of multi-collinearity (VIF < 1.2) among independent variables used. Further, to reduce the effect of location on our results, we included block fixed effects in all regressions.

## 3. Results

### 3.1 Descriptive and summary statistics

**3.1.1. Socioeconomic profile.** A typical farmer (male respondent) was in his 40s with the youngest on average in Mewat (41 years), Haryana and the oldest on average in Bhavnagar (46 years), Gujarat. Farmers in Gujarat (45 years) were significantly older than those in Haryana (43 years). An average farmer household had about seven members in both Gujarat and Haryana. Considering family education, per capita farmer household education (Total years of education of all adult members/number of adult members) was 6.6 years both in Gujarat and Haryana. Approximately 92% of members of a typical farmer household in Haryana and 83% of members of a typical farmer household in Gujarat were engaged in agricultural activities directly or indirectly. As 17% of Gujarat's farmer households were taking part in non-farming occupations, the mean annual farmer household income in Gujarat ($2,655) was much higher than for those farming in Haryana ($514). A lower average income among farmers in Haryana

could be because they largely grow non-cash crops (i.e. wheat and rice) and sell them to government agencies at a minimum support price (MSP, a central government policy in India that ensures a minimum price given to farmers for 24 major crops; [38]). These differences may also be due to inaccurate self-reports of income across both states, with more farmers underreporting their farm incomes in Haryana. More than 70% of income of a farmer household came from farming, with 63% in Gujarat and 83% in Haryana.

**3.1.2. Agricultural profile.** About 95% of farmers in both states were small or medium landholders cultivating less than 10 hectares (ha) of land. More than half of farmers in Gujarat and two-third of farmers in Haryana were small and marginal landholders cultivating less than 2 ha. The average landholding size was 2.6 ha (Table 1), with the largest amount of land owned in Bhavnagar, Gujarat and the smallest amount of land owned on average in Mewat, Haryana. High standard deviation (SD) across all districts shows that landholding distribution was uneven in both states. Although farmers in Haryana cultivated smaller landholding sizes on average (2.1 ha) compared to farmers in Gujarat (3.1 ha), cropping intensity was much higher in Haryana than Gujarat (200% vs 122%) and crops planted were more diversified in Haryana compared to Gujarat (0.62 vs 0.49). In Gujarat, cropping intensity and CDI were estimated as the lowest on average in Bhavnagar (CI: 109; CDI: 0.37) and the highest on average in Banas Kantha (CI: 133%; CDI: 0.63). Farmers on average reported selling 53% of the total crop produced, with the highest in Bhavnagar (60%) and the lowest in Mewat (46%). Considering average distance travelled to purchase food, farmers in Gujarat traveled farther (6.4 kms) than farmers in Haryana (3.2 km) on average. Approximately two-thirds of farmers sold milk in Gujarat whereas only one-thirds of farmers sold milk in Haryana. This is likely due to well-organized and privately managed milk markets in Gujarat.

**3.1.3. Farmer household dietary diversity.** The average FCS (Table 2) of a typical farming household was 67, with the highest in Vadodara (73) and the lowest in Bhiwani (63). The standard deviation of FCS suggests that variation in consumption patterns was higher in Gujarat districts as compared to districts in Haryana. Further, using the World Food Program's categorization of FCS scores [33], all of the farmer households across all six districts in both states were categorized as "Poor", "Borderline" and "Acceptable" (S5 Table) to see their vulnerability in terms of FCS cutoffs defined by WFP [33]. However, the results suggest that most of the farmer households (98%) fell into the "Acceptable" category. Only 1% of farmer households in one district of Gujarat (e.g. Banas Kantha) fell into the "Poor" category though 1–2% of farmer households across all districts in both states were "Borderline" cases. Considering total food groups consumed, on average farming households ate food from eight out of total nine food groups, though there was a variation in this number across districts.

All farming households across all districts in both states consumed main staples, while pulses and vegetables were consumed by all farmer households in Karnal and Mewat districts of Haryana. Most farmer households (97–100%) were consuming milk, sugar, oil and condiments across all districts in both states. Meat and fish was the food group that was consumed by the fewest households (13%) though its consumption was relatively higher in Haryana (21) because Mewat is a Muslim-dominated district where fewer people were vegetarian. Similarly, 14% of farmer households in Vadodara, which is also a Muslim-dominated region, consumed meat and fish. Fruit consumption in Bhiwani (76%) was relatively lower than the respective state (87%) and overall average (90%).

Considering weekly consumption frequency for each of the nine food groups, each farmer household across all districts was consuming staples, milk, sugar, oil and condiments almost each day of the week when our survey was conducted. However, consumption frequencies were lower for fruit, and meat and fish, although they were comparable across all districts in

**Table 2. Farmer household dietary diversity profile of farmer households surveyed in Gujarat and Haryana.**

| Parameter/District/State | Vadodara | Bhavnagar | Banas kantha | Gujrat | Karnal | Bhiwani | Mewat | Haryana | Overall |
|---|---|---|---|---|---|---|---|---|---|
| Food Consumption Score (FCS) and Standard Deviation (in parenthesis) | 73 (10.45) | 71 (8.51) | 64 (9.71) | 69 (10.28) | 65 (8.48) | 63 (7.83) | 65 (6.21) | 64 (7.62) | 67 (9.39) |
| Number of Food groups consumed (#) | 8.0 | 7.9 | 7.8 | 7.9 | 8.0 | 7.7 | 8.4 | 8.0 | 8.0 |
| **Food groups consumed (%) and their consumption frequency (in parenthesis) during the preceding week of survey** | | | | | | | | | |
| Food Group 1: Main staples | 100 (6.7) | 100 (6.9) | 100 (6.7) | 100 (6.8) | 100 (7.0) | 100 (7.0) | 100 (7.0) | 100 (7.0) | 100 (6.9) |
| Food Group 2: Pulses | 99 (5.1) | 99 (4.8) | 96 (3.0) | 98 (4.3) | 99 (3.1) | 99 (2.8) | 100 (2.1) | 99 (2.7) | 99 (3.5) |
| Food Group 3: Vegetables | 99 (5.9) | 99 (5.9) | 99 (5.9) | 99 (5.9) | 100 (5.0) | 99 (4.7) | 100 (5.3) | 100 (5.0) | 99 (5.5) |
| Food Group 4: Fruit | 94 (3.7) | 93 (3.1) | 93 (2.5) | 93 (3.1) | 94 (3.1) | 76 (2.5) | 90 (2.1) | 87 (2.6) | 90 (2.9) |
| Food Group 5: Meat and fish | 14 (1.8) | 2 (1.2) | 2 (1.3) | 6 (1.3) | 8 (1.0) | 2 (1.5) | 53 (1.3) | 21 (1.2) | 13 (1.3) |
| Food Group 6: Milk | 96 (7.0) | 99 (7.0) | 97 (7.0) | 97 (7.0) | 97 (7.0) | 97 (7.0) | 99 (7.0) | 98 (7.0) | 98 (7.0) |
| Food Group 7: Sugar | 99 (7.0) | 100 (7.0) | 98 (7.0) | 99 (7.0) | 99 (7.0) | 100 (7.0) | 100 (7.0) | 100 (7.0) | 100 (7.0) |
| Food Group 8: Oil | 100 (7.0) | 100 (7.0) | 99 (7.0) | 100 (7.0) | 99 (7.0) | 99 (7.0) | 99 (7.0) | 99 (7.0) | 100 (7.0) |
| Food Group 9: Condiments | 100 (7.0) | 99 (7.0) | 99 (7.0) | 99 (7.0) | 100 (7.0) | 97 (7.0) | 99 (7.0) | 99 (7.0) | 99 (7.0) |

both states. Interestingly, the consumption frequency of pulses shows large variations not only across states but also across districts within both states.

## 3.2. Factors associated with FCS

While looking at these associations at the district level (Table 3), the factors associated with FCS vary drastically across regions within both states. For instance, in Gujarat, crop diversity was associated with FCS in Vadodara ($p < 0.05$), whereas, in Bhavnagar, landholding size ($p < 0.05$) was positively associated with FCS while crops sold (%) had a negative association with FCS ($p < 0.05$) here. In Banas Kantha, no factor was associated with FCS.

In Haryana, crop diversity ($p < 0.01$) and landholding size ($p < 0.01$) were significantly associated with FCS in Karnal while in Bhiwani, family education ($p < 0.01$) and crops sold to market ($p < 0.05$) had a significant positive association with FCS. No factor was significantly associated with FCS in Mewat where farmer households had a higher FCS (65) on average compared to the state (64) and consumed the highest number of food groups (8.4 out of 9 against the overall average of 8: Table 2). There was also less variation in FCS across households as evidenced by a smaller standard deviation; this could be due to the fact that the main community of this region (i.e. Muslim) might have followed similar food consumption patterns. To control for sub-district effects on dietary diversity, all of the above district regressions were run with block fixed effects.

## 4. Discussion

Using primary data collected from 1106 households from Gujarat and Haryana, this study investigated the regional differences in which socioeconomic and agricultural factors are associated with household dietary diversity (measured using the FCS) in India. Interestingly, the factors associated with FCS varied across districts in both states. In some regions, agricultural and market factors, such as crop diversity, and percent of crops sold to the market, were the

**Table 3. Regression results showing the agricultural and socioeconomic factors associated with Food Consumption Score (FCS) across districts within Gujarat and Haryana.**

| | Food Consumption Score (FCS) | | | | | |
| --- | --- | --- | --- | --- | --- | --- |
| | Gujarat | | | Haryana | | |
| | **Vadodara** | **Bhavnagar** | **Banas Kantha** | **Karnal** | **Bhiwani** | **Mewat** |
| Crop Diversity Index (CDI) | 0.233** | 0.098 | -0.060 | 0.700*** | -0.098 | 0.154 |
| | (0.095) | (0.081) | (0.085) | (0.205) | (0.167) | (0.121) |
| Landholding Size (in ha) | -0.134 | 0.131** | 0.064 | 0.198*** | 0.141 | 0.040 |
| | (0.103) | (0.066) | (0.083) | (0.071) | (0.102) | (0.145) |
| Per Capita Annual Income (PCAI) | 0.095 | 0.018 | 0.048 | 0.983 | -0.021 | 0.376 |
| | (0.089) | (0.114) | (0.070) | (0.501) | (0.178) | (0.365) |
| Family Education Index (FEI) | 0.151 | -0.064 | 0.118 | 0.040 | 0.233*** | 0.034 |
| | (0.153) | (0.102) | (0.084) | (0.082) | (0.073) | (0.056) |
| Distance to Food Markets (Kms) | -0.009 | 0.102 | 0.053 | 0.268 | 0.101 | 0.087 |
| | (0.123) | (0.061) | (0.067) | (0.153) | (0.085) | (0.109) |
| Crops Sold to Market (%) | 0.024 | -0.191** | 0.116 | -0.203 | 0.239** | -0.028 |
| | (0.106) | (0.086) | (0.090) | (0.104) | (0.102) | (0.074) |
| Milk kept for Domestic use (%) | 0.018 | -0.049 | 0.010 | 0.057 | -0.130 | -0.011 |
| | (0.116) | (0.099) | (0.097) | (0.081) | (0.070) | (0.062) |
| Observations | 106 | 106 | 148 | 153 | 152 | 166 |
| $R^2$ | 0.247 | 0.284 | 0.134 | 0.213 | 0.139 | 0.038 |
| Adjusted $R^2$ | 0.185 | 0.225 | 0.085 | 0.169 | 0.091 | -0.011 |
| Residual Std. Error | 0.997 | 0.788 | 0.813 | 0.843 | 0.770 | 0.687 |
| Sample Size (n) | 98 | 98 | 140 | 145 | 144 | 158 |
| *Block Fixed Effects* | Y | Y | Y | Y | Y | Y |

Significance codes

** $p < 0.05$;

*** $p < 0.01$

factors that were significantly related to FCS. In other regions, socioeconomic factors, such as landholding and education level, were associated with FCS. These results highlight that the factors associated with FCS are diverse and heterogeneous, suggesting that policies to improve household nutrition in rural India may be more effective if they are targeted to specific regions of interest. We will now examine the results for each district in detail, and explain the potential reasons for these differences based on additional information from our survey data, qualitative discussions that we had with farmers in each region, and our knowledge of the social and market context of each district.

First, considering districts in Gujarat, crop diversity (CDI) was the only factor that was significantly associated with FCS in Vadodara. Farmers who planted a more diverse set of crops had a higher household FCS, suggesting that increased crop diversity is associated with improved household nutrition in this region. These results occurred because of the specific cropping and livelihood patterns seen in Vadodara. Vadodara has a relatively high cropping intensity and crop diversity (Cropping intensity = 125%; CDI = 0.48: Table 1) compared to the other two districts of Gujarat in our survey. Yet, the crops that were typically grown were primarily cash crops (66% of the cropped area was under cash crops including cotton, sugarcane and tobacco) that were sold to the market (57% of the crops farmers grew were sold to market: Table 1). In addition, farmers in Vadodara had a larger share of their annual income (38%; Table 1) coming from non-farm sources. The reliance on cash crops and non-farm incomes

may explain why this region had higher annual incomes ($2950; Table 1) compared to farmers from the other two districts in Gujarat. Our results suggest that an increased diversity of crops, primarily cash crops, that are increasingly sold to the market are associated with improved dietary diversity in this relatively prosperous and market-oriented district of Gujarat.

For Bhavnagar (Gujarat), landholding size and the proportion of crops sold to markets were significantly associated with FCS. Previous studies suggest that a larger landholding size can improve household dietary diversity [9,34] by increasing income, which may allow farmers to buy more diverse food from markets [12,14]. In addition, by having larger landholdings, farmers were able to plant a more diverse range of crops on farm (there is a positive significant relationship between landholding size and crop diversity in this district, $p < 0.01$). Interestingly, the association between the proportion of crops sold to markets and FCS was opposite of what we hypothesized; in this region, farmers who sell more crops to the market are more likely to have lower FCS. It is possible that farming households that sell a higher proportion of their crops to market are more dependent on purchased food for consumption. If local markets do not commonly offer diverse foods, or if these foods are too expensive for low-income households, dependency on market purchased foods may limit dietary diversity [34]. In Bhavnagar, the consumption of several diverse food items (e.g., fruits, meat and fish; Table 2) that are not commonly cultivated or reared, but rather are purchased, was low. As per our discussions with farmers in this area, the availability of meat items from local markets was limited in Bhavnagar and if they were available, they were very expensive because most non-Muslim communities in this region prefer eating vegetarian food.

In Banas Kantha, Gujarat, no factors considered in our study were significantly associated with FCS. Relatively, the adjusted $R^2$ of the linear regression for this region was also much smaller than the adjusted $R^2$ of the other district models in Gujarat (0.085; Table 3). This suggests that we may not have captured the factors in our survey that could be more important in explaining dietary diversity in Banas Kantha because this region appears to be least reliant on farming for their livelihoods, with only 56% of the average farmer's income coming from farming (Table 1). It is therefore possible that there are non-farm factors that are important in explaining FCS, such as consumer behavior and the availability of diverse food items from markets, that we did not consider in our study.

Considering districts in Haryana, in Karnal, crop diversity and landholding size were positively and significantly associated with FCS. The importance of crop diversity in this region may be because households here were heavily dependent on farming, with 94% of family members engaged in farming and with 87% of annual income coming from farming (Table 1). Crop diversity may be particularly important in this region given that the majority of food grain crops grown in this area are procured on assured price (i.e. MSP) by the government, and there is subsequently less crop diversity in this region compared to other two districts in Haryana. Given farmers' reliance on farming as a primary livelihood, and given that there is overall low crop diversity, it is possible that those farmers who grow more diverse crops on farm end up having higher dietary diversity. In addition, large landholders have more crop production, resulting in increased crop income, and they also are able to plant more diverse crops since they do not have limitations due to land. This may result in more diverse diets both through increased diversity of foods produced on farm and more income to purchase foods from the market.

In Bhiwani, Haryana, increased levels of education and selling more crops to the market were associated with improved FCS. This may be because Bhiwani is the poorest district of all of the ones considered in our study, with households on average earning $466 per year. In addition, this community is heavily dependent on farming, with 86% of annual income coming from farming (Table 1). Finally, this is also the district where households consumed the

least amount of fruit, with only 76% of households consuming fruit 2.5 times in the preceding week of survey (Table 2). In this region, through discussions with farmers, we found that most fruit that households consumed was purchased from market. Therefore, in this community that is heavily-dependent on farming, earning income from selling crops may be the most viable way to access additional food groups, such as fruit, that must be purchased from markets.

Finally, in Mewat (Haryana), no variables in our regressions were significantly associated with FCS, and the adjusted $R^2$ of our model was negative (-0.011), the lowest across all districts (Table 3). Our inability to detect any significant factors may be because the overall FCS was high in this region (farmer households here consumed 8.4 out of total 9 food groups in the preceding week of survey) and had low variation (*Mean* = 65; *SD* = 6.21; Table 2). This lack of variation in FCS may be because most members of the community are Muslims and may have similar eating patterns; specifically, in this region eating non-vegetarian food items is common and ultimately increases FCS. In addition, similar to Banas Kantha, farmers earned the least amount of income from farming in this district (77% of income came from farming) compared to the other districts in Haryana. This suggests that there may be some non-agricultural variables that are important explanatory factors for FCS in this less farming-dependent region that we did not consider in our study.

Overall, our findings suggest that farmer household dietary diversity in rural India is associated with a variety of agricultural and socioeconomic factors. Importantly, we find that which factors are significantly associated with FCS vary across districts even within the same state, suggesting that the drivers of dietary diversity may be region-specific. Our results suggest that future agricultural and socioeconomic policies that aim to increase dietary diversity in India may benefit from considering regional differences in the drivers of dietary diversity associated with social and economic contexts. These policies should consider the reality of the farming communities on the ground, and understand that there are heterogeneous drivers of decision-making and food access based on the location where a farming household lives.

It is important to note that this study has several limitations. First, some of our results are correlational and based on observation data, and therefore are not causal. Future work should follow the same farmers through time to collect panel data to better understand how changes in agricultural and socioeconomic variables within a given household influence changes in dietary diversity. In addition, we only collected dietary data from one time point within a year. Specifically, we focused our survey during the summer season when farmers are less likely to be growing crops and therefore may be more food insecure. Future work should examine how these results vary if dietary information is collected in different seasons across a year. Finally, we did not collect data on how much of the food that the respondents consumed was home grown or purchased from market. This would have allowed us to better understand whether farmers' dietary diversity was being driven more by what was grown on farm or through income generation that allowed farmers to purchase diverse food from market. Future work should tease apart this important distinction to better understand the mechanisms underpinning the associations seen in this study.

## 5. Conclusions

Using primary data collected from 1106 farmer households in India, this paper investigated the regional differences in socioeconomic (family education, income, distance to food markets, and milk kept for domestic purpose) and agricultural (crop diversity, landholding size, crops sold to market) factors associated with farmer household dietary diversity in India. Our results found that the factors associated with FCS were region-specific across and within each of the two states. In some regions, higher crop diversity and better education could improve farmer

household dietary diversity, and in other regions, dietary diversity is best improved through income generation where farmers were able to purchase more diverse foods from markets using the income they earned from selling their crops to market. In conclusion, our results highlight that the factors associated with improved household nutrition are heterogeneous and vary across regions even within the same state. This suggests that future agricultural and socio-economic policies should consider regional-level variation in the drivers of household dietary diversity by understanding local social and economic contexts.

## Supporting information

**S1 Table. Calculation of a district-wise Farming Intensity Index (FII) for Gujarat.**
(DOCX)

**S2 Table. Calculation of a district-wise Farming Intensity Index (FII) for Haryana.**
(DOCX)

**S3 Table. Calculation of a block-wise Farming Intensity Index (FII) for districts in Gujarat.**
(DOCX)

**S4 Table. Calculation of a block-wise Farming Intensity Index (FII) for districts in Haryana.**
(DOCX)

**S5 Table. Number of farmer households falling under "Poor", "Borderline" or "Acceptable" FCS categories.** *
(DOCX)

## Acknowledgments

We are indebted to Professor William Masters (Tufts University, Boston, USA) for his help and guidance during conceptualizing and designing the research methodology for this study. We are also thankful to our colleagues, Dr. Preeti Rao and Dr. Nishan Bhattarai (University of Michigan, Ann Arbor, USA) for their valuable inputs at various stages of this project. Last but not the least, we are sincerely grateful to respondents who spared so much time to participate in the field survey, and enumerators who helped us collect data during a hot summer season.

## Author Contributions

**Conceptualization:** Sukhwinder Singh, Meha Jain.

**Data curation:** Sukhwinder Singh.

**Formal analysis:** Sukhwinder Singh.

**Funding acquisition:** Sukhwinder Singh.

**Investigation:** Sukhwinder Singh.

**Methodology:** Sukhwinder Singh, Andrew D. Jones, Meha Jain.

**Project administration:** Sukhwinder Singh, Meha Jain.

**Resources:** Sukhwinder Singh.

**Software:** Sukhwinder Singh.

**Supervision:** Meha Jain.

**Writing – original draft:** Sukhwinder Singh.

**Writing – review & editing:** Sukhwinder Singh, Andrew D. Jones, Meha Jain.

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
