## [Decision Letter · Decision Letter 0]

2 Jan 2020

PONE-D-19-31872

Regional differences in agricultural and socioeconomic factors associated with farmer household dietary diversity in India

PLOS ONE

Dear Dr Singh,

Thank you for submitting your manuscript to PLOS ONE. After careful consideration, we feel that it has merit but does not fully meet PLOS ONE’s publication criteria as it currently stands. Therefore, we invite you to submit a revised version of the manuscript that addresses the points raised during the review process.

We would appreciate receiving your revised manuscript by Feb 16 2020 11:59PM. To enhance the reproducibility of your results, we recommend that if applicable you deposit your laboratory protocols in protocols.io, where a protocol can be assigned its own identifier (DOI) such that it can be cited independently in the future. For instructions see: http://journals.plos.org/plosone/s/submission-guidelines#loc-laboratory-protocols

We look forward to receiving your revised manuscript.

Kind regards,

William Joe

Academic Editor

PLOS ONE

Journal Requirements:

2. Thank you for including your ethics statement in the manuscript: 'This survey was reviewed and approved by the Institution Review Board (IRB Approval Number: IRB00000246).'

NO

We are indebted to the IMMANA Fellowship for providing funding [...] The study was conducted under a Postdoctoral Fellowship Program, “Innovative Methods and Metrics for Agriculture and Nutrition Actions  (IMMANA)” funded by the UK Department for International Development ( 372 DFID), coordinated by The Leverhulme Centre for Integrative Research on Agriculture and Health (LCIRAH), and managed by Tufts University, Boston, USA (Grant number: IMMANA 169864-2013).

Sukhwinder Singh

Grant number: IMMANA 169864-2013

the UK Department for International Development (DFID)

https://immana.lcirah.ac.uk

No

Reviewers' comments:

Reviewer's Responses to Questions

**Comments to the Author**

1. Is the manuscript technically sound, and do the data support the conclusions?

Reviewer #1: Yes

Reviewer #2: Partly

2. Has the statistical analysis been performed appropriately and rigorously? 

Reviewer #1: Yes

Reviewer #2: No

3. Have the authors made all data underlying the findings in their manuscript fully available?

Reviewer #1: Yes

Reviewer #2: Yes

4. Is the manuscript presented in an intelligible fashion and written in standard English?

Reviewer #1: Yes

Reviewer #2: Yes

5. Review Comments to the Author

Reviewer #1: The comments on the paper entitled “Regional differences in agricultural and socioeconomic factors associated with farmer household dietary diversity in India” are as follows:

This study tried to understand the regional-level differences in factors associated with farmer household dietary diversity in the states of Haryana and Gujarat in the country. Household dietary diversity is measured by using the Food Consumption Score (FCS). The survey data collected from 1106 households is employed to analyze the objectives of the study. The determinants of household dietary diversity are analyzed through multiple regression models. The model is checked for multi-collinearity by ensuring that VIF values were less than 3. Block fixed effects, as a robustness check, was also included in all the regressions to reduce the effect of individual sites on the results.

The paper is fairly-written, the variables selected for analysis are explained in detail in the methodology section, and the interpretation and discussion of the regression results are fine. The limitations of the study are clearly brought out in the text. I find this paper is fit for publication in the journal. However, author(s) may consider the following minor suggestions:

1. In the light of results of the study, author(s) may suggest some policy measures in the last section for both Haryana and Gujarat. It would enrich the quality of the paper.

2. Please verify the following sentence in the text: all of our results are correlational and based on observation data, and therefore are not causal as mentioned in the last paragraph of page no 15. Study has used regression analysis to understand the determinants, hence the phrase ‘all of our results’ should be replaced with ‘some of our result’.

Reviewer #2: This paper investigated whether there are any regional-level differences in factors associated with farmer household dietary diversity. The paper particularly focused on the dietary diversity of farmer households which has been infrequently studied in India. However, the authors have failed to explain how they define farmer households. Whether their consideration was based on possession of the land or income-earning from farming activities is not clear. What about the tenants – whether they have been included.

This paper has applied the Food Consumption Score (FSC) approach developed by the World Food Programme (2008) to estimate household dietary diversity. The authors have presented a summary of FSC calculated in the paper and discussed its spatial pattern. But it would have been more meaningful if they have classified households as having "poor," "borderline," or "acceptable" food consumption by applying the WFP's recommended cut-offs to the food consumption score. This could provide information on households' food consumption status and which region/state/district is vulnerable. The generated categorical variable could further be used as the response variable in regression analyses, which may give a different result from the present one.

In this paper, primary data collected from two Indian states - Gujarat and Haryana have been analyzed. The authors discussed the sampling strategies adopted in the paper. However, the justification for the selection of the above states was discussed loosely. Both states are relatively small in size but share different geographical and farming characteristics and food habits as well.

The paper used linear regression methods to identify the associations between agricultural (e.g. crop diversity, landholding size, farm income and crops sold) and socioeconomic (e.g. family income, family education, distance to food markets and consumption of domestically produced milk) factors, and FCS. Results presented in Table 3 show drastic variation in parameters across different districts. This variation could be occurred due to change in regions, highlighting region-specific policy focus. But there is a possibility that the variation in parameters could be due to the presence of multicollinearity among some predictors. Landholding size, Per Capita Annual Income, and crops sold to the markets seem to be highly correlated with each other.

The authors have also ignored few important socio-economic variables like caste and region. Such variables are critical in explaining households’ dietary pattern in India. Similarly, policy variables like access to public distribution system have also been ignored. Some variables based on farm policies of these states should have also been considered for better policy implications at regional level.

In summary, the theme discussed in the paper is interesting - particularly in terms of providing useful inputs to farm and health policy. However, the authors have left several scopes for further improvements of the paper. Therefore, the current version of the manuscript could not be approved.

6. PLOS authors have the option to publish the peer review history of their article (what does this mean?). If published, this will include your full peer review and any attached files.

Reviewer #1: No

Reviewer #2: Yes: Amarnath Tripathi

---

## [Author Response · Author response to Decision Letter 0]

14 Feb 2020

Dear Reviewers

We are sincerely thankful for providing very useful feedback. We tried to make all the suggested changes and answered your comments. Please find attached the updated versions of the manuscript (with and without track changes), a document detailing our answers to your comments and suggestions, and an updated version of supplementary information.

Specific grant numbers: IMMANA 169864-2013

Initials of authors who received each award: Sukhwinder Singh

Full names of commercial companies that funded the study or authors: The UK Department for International Development (DFID)

Initials of authors who received salary or other funding from commercial companies: Sukhwinder Singh

URLs to sponsors’ websites: https://immana.lcirah.ac.uk

---

## [Decision Letter · Decision Letter 1]

17 Mar 2020

Regional differences in agricultural and socioeconomic factors associated with farmer household dietary diversity in India

PONE-D-19-31872R1

Dear Dr. Singh,

We are pleased to inform you that your manuscript has been judged scientifically suitable for publication and will be formally accepted for publication once it complies with all outstanding technical requirements.

With kind regards,

William Joe

Academic Editor

PLOS ONE

Additional Editor Comments (optional):

Reviewers' comments:

Reviewer's Responses to Questions

**Comments to the Author**

1. If the authors have adequately addressed your comments raised in a previous round of review and you feel that this manuscript is now acceptable for publication, you may indicate that here to bypass the “Comments to the Author” section, enter your conflict of interest statement in the “Confidential to Editor” section, and submit your "Accept" recommendation.

Reviewer #2: All comments have been addressed

2. Is the manuscript technically sound, and do the data support the conclusions?

Reviewer #2: Yes

3. Has the statistical analysis been performed appropriately and rigorously? 

Reviewer #2: Yes

4. Have the authors made all data underlying the findings in their manuscript fully available?

Reviewer #2: Yes

5. Is the manuscript presented in an intelligible fashion and written in standard English?

Reviewer #2: Yes

6. Review Comments to the Author

Reviewer #2: (No Response)

7. PLOS authors have the option to publish the peer review history of their article (what does this mean?). If published, this will include your full peer review and any attached files.

Reviewer #2: Yes: Amarnath Tripathi